# Investigating the Retrofitting Effect of Fiber-Reinforced Plastic and Steel Mesh Casting on Unreinforced Masonry Walls

**DOI:** 10.3390/ma16010257

**Published:** 2022-12-27

**Authors:** Faizan Halim, Afnan Ahmad, Mohammad Adil, Asad Khan, Mohamed Ghareeb, Majed Alzara, Sayed M. Eldin, Fahad Alsharari, Ahmed M. Yosri

**Affiliations:** 1Department of Civil Engineering, CECOS University of Science and Emerging Sciences Peshawar, Peshawar 25120, Pakistan; 2Department of Civil and Environmental Engineering, University Teknologi Petronas, Seri Iskandar 32610, Malaysia; 3Department of Civil Engineering, University of Engineering and Technology Peshawar, Peshawar 25000, Pakistan; 4Department of Civil Engineering, College of Engineering, Jouf University, Sakaka 72388, Saudi Arabia; 5Center of Research, Faculty of Engineering, Future University in Egypt, New Cairo 11835, Egypt

**Keywords:** unreinforced masonry (URM), fiber-reinforced, steel mesh, out of plane bending, diagonal tension, compressive test, retrofitting

## Abstract

Unreinforced masonry (URM) is one of the most popular construction materials around the world, but vulnerable during earthquakes. Due to its brittle nature, the URM structures may lead to a possible collapse of the wall of a building during earthquake events causing casualties. In the current research, an attempt is made to enhance the seismic capacity of URM structures by proposing a new innovative composite material that can improve the shear strength and deformation capacity of the URM wall systems. The results revealed that the fiber-reinforced plastic having high tensile and shear stiffness can significantly increase in-plane as well as out-of-plane bending strength of the URM wall. It was recorded that the bending moment of the prism increased up to 549.5% by increasing the bending moment from 490 N*mm to 3183 N*mm per mm deflection of prism upon using glass fibers. Moreover, the ductility ratio amplified up to 5.73 times while the stiffness ratio increased up to 4.16 times with the aid of glass fibers. Since the material used in this research work is low cost, easily available, and no need for any skilled labor, which is economically good. Therefore, the URM walls retrofitted with fiber-reinforced plastic is an economical solution.

## 1. Introduction

Unreinforced masonry (URM) structures are the most favorable way of construction for human beings since far-off ages due to their resilience to environmental effects, durability as well as ease of construction. It can be seen around the world in various forms which mainly depend upon construction material availability, in different forms around the globe depending upon the availability of construction materials, the monetary condition of the area, and its local culture [1]. Placing pebbles, bricks, or blocks one above the other using any binding agent, such as mortar, is the easiest form. Blocks of concrete are the most contemporary type of construction unit, while the stone is considered the first unit used in the construction of masonry structures pursued by sun-dried and burnt clay bricks.

Nowadays, the utmost number of unreinforced masonry structures are constructed without considering any seismic resistance features, especially in rural areas [2]. On the other side, it has been observed from the recent earthquake around the globe that masonry structures are highly seismic vulnerable [3]. A devastating earthquake having a magnitude of 7.6 caused more than 73,000 fatalities, injuring about 70,000 local residents and ultimately affecting 3.5 million people in the northwest region of Pakistan, specifically Kashmir [4]. The Asian Development Bank and World Bank Report [ADB-WB 2005] show that the earthquake devastated around 203,579 residential units and damaged 196,574 other facilities [5]. Since the damaged buildings suffered during the earthquake consist of stone and brick masonry constructions [6]. Therefore, it is necessary to improve the seismic performance of unreinforced masonry buildings by retrofitting or rehabilitation.

An earthquake triggers various types of structural damage in masonry structures [7]. Various aspect of different types of damages generated by an earthquake in a masonry structure has been summarized in a technical report prepared by The Applied Technology Council [8]. Reconstruction of the affected facility due to an earthquake is very expensive, causing substantial economic loss. For this reason, retrofitting the affected building is considered the most economical solution and recommended alternative to mitigating the associated seismic risk [9]. Several conventional techniques, such as ferrocement overlay [10], grout injection [11], reinforced concrete jacketing [12], application of steel elements [13], bed joint reinforcement [14], polypropylene band [15], etc., and modern techniques such as the application of fiber-reinforced plastic [16], center core technique [17], and post-tensioning [18] are currently being used around the globe for retrofitting and rehabilitation of masonry structures [19]. However, non-conventional techniques are costly and require special materials and technologies [20]. To cope with this situation, a thorough literature survey about several retrofitting methods, and glass fiber retrofitting techniques were selected for this particular research. The availability of local materials and skilled labor was one of the main criteria for selecting this technique. This study focuses on comparing unreinforced masonry structures retrofitted with glass fibers. The primary objective of the research study is to investigate the strength, stiffness, and durability of unreinforced masonry walls utilizing glass fiber retrofitting techniques.

Several research works have investigated the stability of unreinforced masonry walls and retrofitted masonry walls under static and dynamic loading conditions [21]. Elgawady et al. [22] performed experimentation on unreinforced masonry walls, which are retrofitted with shotcrete overlay. Three prototype masonry walls by means of half-scale clay bricks and the required amount of mortar were made. Keeping one of them as a controlled unit, the performance of shotcrete layer thickness has been assessed using the remaining two walls by retrofitting one wall with a single-sided 40 mm dense shotcrete layer and providing a 30 mm layer on both sides of the other wall. The results reveal that the lateral load strength of the tested wall increased by three while providing shotcrete layers on both sides, enhancing the ductile failure. Adopting the same technique, Lawrence F. Kahn [23] prepared fifteen three square feet (91.44 cm × 91.44 cm) and four square feet (121.92 cm × 121.92 cm) brick masonry panels and retrofitted them with 3.5 inches of reinforced shotcrete layer. Exposing the panels to the diagonal tension test, he observed a 6 to 25 times increased ultimate load. Sivaraj et al. [24] experimentally investigated the out-of-plane shear behavior of two types of masonry wall: a controlled burned clay brick masonry wall strengthened with glass fiber-reinforced polymers (GFRP) glued externally. It has been observed from the failure mode on the wall that GFRP reasonably controls the impact of an earthquake. Thus, several attempts have been made to control the damage to the masonry walls; however, the seismic performance of the URM still needs proper attention. This research focuses on the shear strength and deformation capacity of the URM wall systems retrofitted with glass fiber and ferrocement strips. For this purpose, a prism of a specific size is constructed and tested through compressive strength, flexural strength, and diagonal tension tests. The results obtained indicated significant improvement of the URM retrofitted walls, highlighting the applications of glass fiber and ferrocement.

## 2. Experimental Section

### 2.1. Preparing Prisms for Testing

All the casted prisms have dimension of 228.6 mm thickness, 609.6 mm height, and 609.6 mm length. A motor paste was prepared using 1:4 (cement:sand). A wooden base having a dimension equal to 762 mm × 304.8 mm (length × width) has been used to construct a prism. First-class bricks and the above-mentioned materials were used to construct prism adopting the English bond of bricks.

#### 2.1.1. Glass Fiber Sample

Seven samples out of twenty-one samples were coated with glass fiber. Glass fiber sheet was prepared locally and then cut into strips of dimension 609.6 mm (2ft) in length and 50.8 mm (2 inches) wide and then fixed on the wall with screw and steel washer and also the specified bonding solution was sprayed before fixing the strips for a proper joint with the wall as shown in Figure 1a.

These samples were fitted so that a 50.8 mm gap was fixed between each successive strip; also, at the ends of the wall (228.6 mm thick), glass fiber strip was coated for continuity purposes as shown in Figure 1a.

#### 2.1.2. Ferrocement Strip Samples Casting

Out of twenty-one, seven samples were plastered with wire mesh used (dimension). The wire mesh was cut in 50.8 mm thickness and 304.8 mm in length, as shown in Figure 1b, and then was fitted on brick masonry with screw and steel washer in such a way that each strip was fixed on the wall with 50.8 mm gap between two successive strips. These samples were then plastered in 1:3 (cement:sand) and then cured.

### 2.2. Testing of Brick and Prisms

#### 2.2.1. Compressive Strength Test of Brick

The compressive load-bearing capacity of bricks used in the tested prism has been investigated, following the guidelines of ASTM C67 [25]. According to the prescription, capping has been provided using cement sand mortar to fill frogged bricks and allow uniform loading. A universal testing machine (UTM) (UH-200A Shimadzu, Kyoto, Japan) was utilized to investigate the compression capacity of bricks by applying the compression load on the bed (228.6 × 228.6 mm^2^) of the brick.

#### 2.2.2. Prism Flexure Test

The out-of-plane strength of the prism has been investigated following the guidelines of ASTMC78/C78-M18 [26]. Six samples having dimensions 0.609 m × 0.609 m (2ft × 2ft) with 228.6 mm (9 inches) thick prism were tested in flexure. A controlled sample and two prisms retrofitted with glass fiber retrofitted were tested. The test was conducted in such a way that the prism was supported by two steel rods, each having a 50.8 mm (2 inches) diameter and 609.6 mm (2 ft) length separated at a 508 mm (20 inches) distance. The vertical force was applied at the center of a steel road placed in the middle of a prism using a hydraulic loading jack having a capacity of 200 kN (20 tons) as shown in Figure 2a. Rubber pads having a thickness equal to 12.7 (0.5 inches) were placed beneath each steel rod to ensure uniform distribution of stresses, avoiding stress concentration and local failure.

The testing was carried out in a movable straining frame allowing movement in the horizontal direction and applying vertical load through hydraulic jacks. Firstly, the prism was placed on supports of steel rods while the hydraulic jacks were set on the center of the prism. In order to record the real-time load, a data logger (UCAM-70A; Kyowa, Japan) was connected to the calibrated cell as well as a displacement sensor was used to record the vertical displacement of the prism, as shown in Figure 2b.

#### 2.2.3. Prism Diagonal Compression Test

This test was carried out according to the ASTM E519/E510-10 [27]. The objective/goal of the test was to find out the tensile capacity, shear stress, shear strain, and shear modulus of the samples. The dimension of the sample that was used was 609.6 mm in length, 609.6 mm in width, and 2743.2 mm in thickness. A total of 9 specimens were made and then 6 of them were retrofitted because of the local failure in the specimens. 

To record the horizontal lengthening and vertical shortening of the prism two displacement sensors were installed in the diagonal direction having an average gauge length of 490 mm. Load cell was also installed at the top of the shoe of the sample. Load cell and displacement sensors (Kanetec Peacock, Japan) were then connected to the data logger for the data record. Vertical load causes shear stress in the sample. The diagonal tension test setup is shown in Figure 3.

#### 2.2.4. Compressive Strength Test of Prism

The compression test of samples was carried out in accordance with ASTM E447-74 [28]. The dimension of the compression specimen was 609.6 mm in length, 609.6 mm in width, and 2743.2 mm in thickness. A total of three compression samples were constructed. All compression specimens were tested after 28 days. The compression specimens of samples were tested under uni-axial compressive loading till failure as shown in Figure 4a,b. The compressive strength for all compression specimens of samples’ compressive stresses was calculated by dividing the peak load by the area exposed to the load.

Compression test of samples was carried out in universal testing machine (UTM) having 200 tons load capacity. Two displacement sensors were installed, one in the front of the specimen and one in the back of the compression specimen of samples. The compressive strength of masonry was calculated by dividing peak vertical load by the cross-sectional area of a prism. Modulus of elasticity was calculated from the stress–strain curve as a secant modulus between two points corresponding to 1/20th and 1/3rd of the peak vertical stress.

## 3. Results and Discussion

### 3.1. Brick Compression Strength

The compression capacity of three days cured bricks used in the construction of the test prism is illustrated in Figure 5. It can be seen in Table 1 that the mean strength of three randomly taken bricks is about 12,064.37 kN/m^2^. Moreover, it has been learned that the average strength of middle to high-class bricks ranges from 10,342.14–13,789.51 kN/m^2^ [29]. Therefore, it can be stated that middle to high-class bricks was used in the assessment of unreinforced masonry wall retrofitting using glass fibers.

### 3.2. Prism Flexural Strength Adopting Bilinear Idealization Approach

Load displacement bilinear idealization is based on the envelope curve of load–displacement response, identifying the transition behavior of a structure from elastic to inelastic deformation [30]. Reviewing the existing idealized bilinear methods for various nonlinear testing data interpretation procedures [31], the author decided to utilize it for the unreinforced masonry prism retrofitted by glass fiber. Figure 6 illustrates the bilinear idealization of an elasto-plastic behavior of a structure, assuming that area under the bilinear curve is equivalent to the area below the curve.

Additionally, the straight line between 0.0 to 0.4 P_u_ defines initial stiffness while the maximum deformation (post-peak displacement) occurred at a 20% reduction of P_u_. Moreover, P_y_ (yield load) can be obtained using Equation (1) [32].
(1)$$P_{y}=[\Delta_{\mathrm{failure}}-\sqrt{\Delta_{{\mathrm{failure}}^{2}}-\frac{2\times w_{failure}}{K}}]\times K$$
where P_y_ is the yield load, Δ_failure_ represents the deformation at failure, $w_{failure}$ which indicates the energy dissipated until failure occurs. 

The load-taking capacity, as well as the bending moment of the prism using glass fiber, were assessed. The bending moment capacity of two tested prisms is shown in Figure 7. It can be observed that one prism (GF1) attains a maximum bending moment of 3166.67 N-mm with a maximum deformation of 11.55 mm. While the other prism (GF2) reaches the highest value of bending moment, up to 3184.72 N-mm with deformation of 8.98 mm. According to the Australian masonry code (AS3700), the permissible flexural strength of the unreinforced masonry wall is about 200 N-mm. It can be observed that both the prism retrofitted with glass fiber satisfies the minimum permissible strength criteria due to the higher bending capacity caused by the glass fiber confinement [33]. Similar studies have been reported by Messali et al. [34]. Moreover, Kalali and Kabir [35] have proved that glass fiber significantly reduces the seismic risk of the masonry wall by increasing the bending capacity and ductility.

Figure 8 illustrates the load–displacement curve of the two tested prisms and their corresponding bilinear idealized curves. It is evident from the idealized bilinear curve that the glass fiber retrofitted prism possesses considerable load-bearing capacity in inelastic deformation zone. It indicates that glass fiber increases the ductile behavior of the unreinforced masonry wall (URM). D.P. Abrams [36] claimed similar findings for the unreinforced masonry walls. All the characteristics and parameters of the idealized bilinear envelope are shown in Table 2. 

### 3.3. Stiffness and Ductility of Prisms

Stiffness and ductility are the two parameters often used to assess the out-of-plane failure of unreinforced masonry walls [37]. Stiffness indicates the ability of a structure to withstand external loading in an elastic region, while ductility reflects the ability of a structure to experience inelastic deformation without significant loss is worth deriving [38]. Ductility can be defined in many ways depending on the parameter under consideration. According to the definition of ductility stated in ASTM E-2126, the ratio of ultimate displacement (Δ_u_) at 80% of the peak load to the displacement (Δ_yield_) at yield limit state (P_yield_) [39]. It can be expressed mathematically by Equation (2).
(2)Du=ΔfailureΔyeild

Comparing the stiffness of the control prism coated with ferrocement and glass fiber, it can be witnessed from Figure 9 that the stiffness ratio increased up to 12.93 times using ferrocement and 4.16 times using glass fiber. The lower stiffness of the glass fiber retrofitted prism compared to ferrocement is due to the stretchable nature of the glass fiber strips yielding higher ultimate strength as evidenced by Kalali and Kabir [35], while the higher stiffness ferrocement-coated prism indicates brittle nature depreciated in earthquake-prone zones [40].

The ductility ratio of the glass fiber-coated prism is 5.73 times greater than the controlled one, while the ferrocement-coated prism enhances ductility up to 0.75 times, as shown in Figure 10. The higher ductility exhibited by the glass fiber is due to its stretchable nature as evidenced by Kalali and Kabir [35]. Therefore, the enhanced results of using glass fiber coating have proven its feasibility in retrofitting unreinforced masonry walls. 

### 3.4. Elastic Modulus (E) of Prisms

The elastic modulus is a ratio between stress and corresponding strain while the stress should be below the proportional limit. It measures the rigidity and stiffness of a material. In terms of the stress–strain curve, the slope of the curve in the range of proportional limit is the modulus of elasticity.

The ASTM E447-74 [28] was used to evaluate the elastic modulus, and the secant modulus was utilized to determine the value based on the peak vertical stress of 1/20th and 1/3 peak. Figure 11 and Table 3 present the results of the controlled sample (CS), prism coated with glass fiber (GF), and ferrocement (FCS). The modulus of elasticity of the CS is 1.64 MPa, while that of GF and FCS is 2.28 MPa and 2.48 MPa. The percent increase in the modulus of elasticity of FCS and GF as compared to CS is 71.45% and 59.68%, respectively. The confining effect caused by the ferrocement and anchorage properties provided by the glass fiber noticeably increases the elastic properties of the coated prism [41].

### 3.5. Shear Modulus (G) of Prisms

The shear modulus, often denoted by G is a measurement of the prism’s rigidity and it is calculated from the ratio of shear stress to shear strain.

The shear strength, corresponding strain, and shear modulus are presented in Table 4. The shear modulus of CS is given as 275 MPa, while that of GF and FCS is 227.3 MPa and 393.86 MPa. The shear modulus values were obtained from the stress strain curves shown in Figure 12, Figure 13 and Figure 14. The percent increase in the shear modulus of FCS as compared to CS was 43.22%, while a 16.25% decrease was observed in GF. The increase in the shear modulus of FCS is attributed to the combined effect of steel fiber and ferrocement overlay. 

## 4. Conclusions

The feasibility of utilizing glass fiber as a means of retrofitting material in unreinforced brick masonry (URM) walls has been investigated in this research. The flexural strength of the brick’s prism was investigated along with the ductility and stiffness ratio. The following conclusions can be stated from the results.
A significant increase has been observed in the bending moment of prism utilizing glass fiber. It can be seen that the bending moment of a controlled specimen is about 490 N*mm/mm while it raised to 3183 N*mm/mm (average bending moment of two prisms) by means of glass fibers.The ultimate load-bearing capacity of a prism using glass fibers has been increased considerably along with the ductility and stiffness ratios. It can be observed that the ductility ratio amplified up to 5.73 times while the stiffness ratio increased up to 4.16 times with the aid of glass fibers.It has been observed from the failure pattern that glass fibers de-nailed firstly followed by the cracking of the brick prism.

Based on the results obtained, it is clear that retrofitting with glass fiber and ferrocement significantly enhances the performance of the URM which is an economical solution. In addition, it is recommended for future work to assess the seismic performance of the URM walls retrofitted with glass fiber and ferrocement.

## Figures and Tables

**Figure 1 materials-16-00257-f001:**
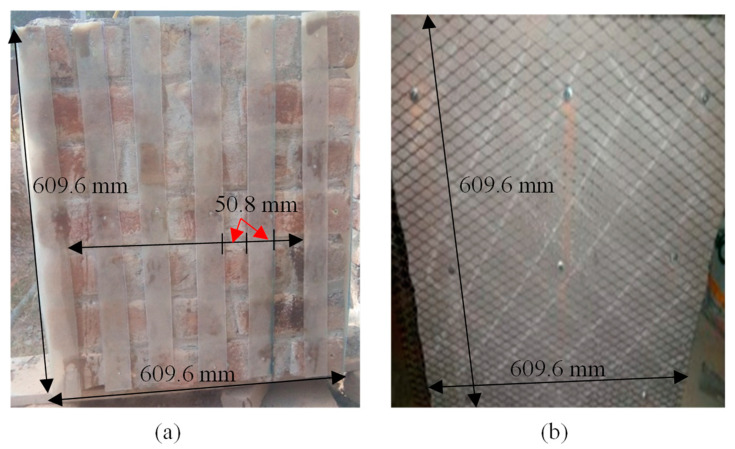
(**a**) Glass fiber strip and (**b**) ferrocement arrangement.

**Figure 2 materials-16-00257-f002:**
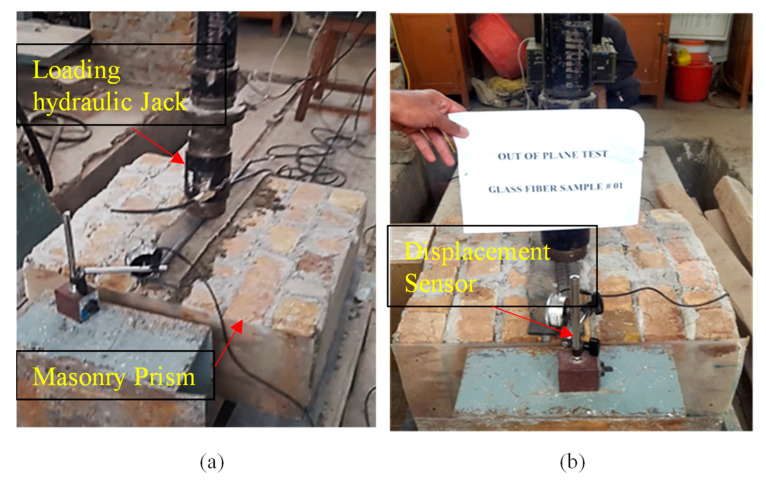
Experimental setup for the flexural strength test (**a**) loading setup over prism and (**b**) data logger setup.

**Figure 3 materials-16-00257-f003:**
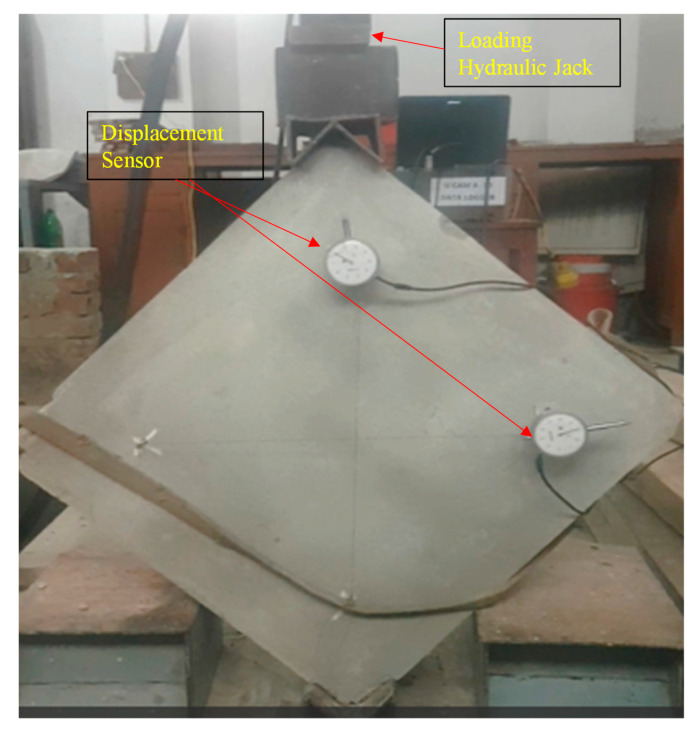
Experimentation of diagonal tension test.

**Figure 4 materials-16-00257-f004:**
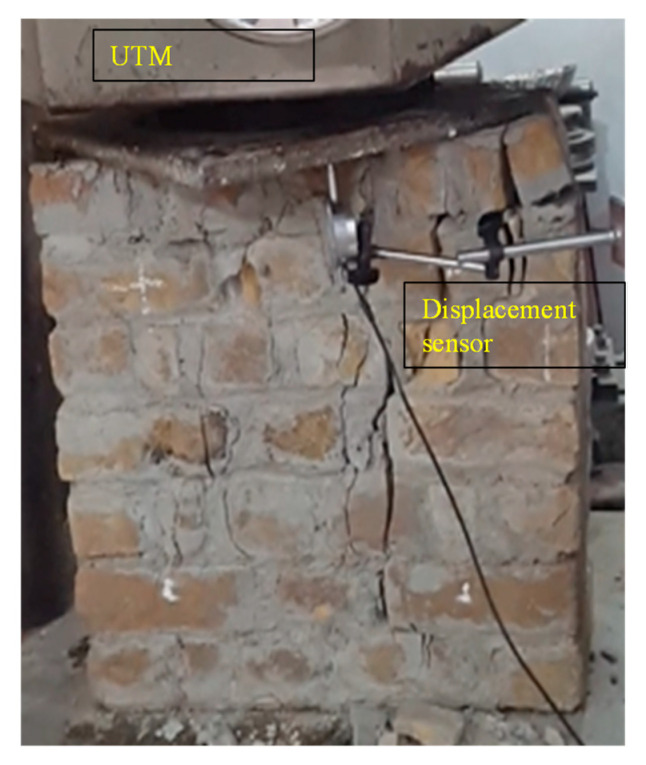
Experimental setup for the compressive strength of prism.

**Figure 5 materials-16-00257-f005:**
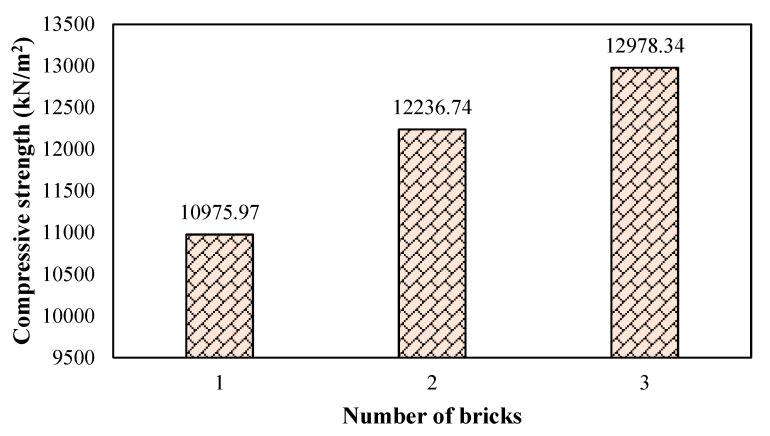
Compression capacity of bricks used in the test prism.

**Figure 6 materials-16-00257-f006:**
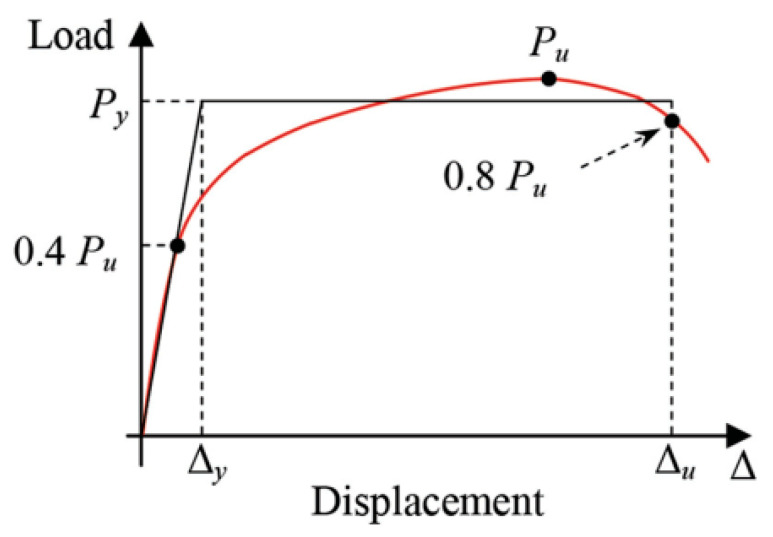
Equivalent energy elastoplastic bilinear idealization method.

**Figure 7 materials-16-00257-f007:**
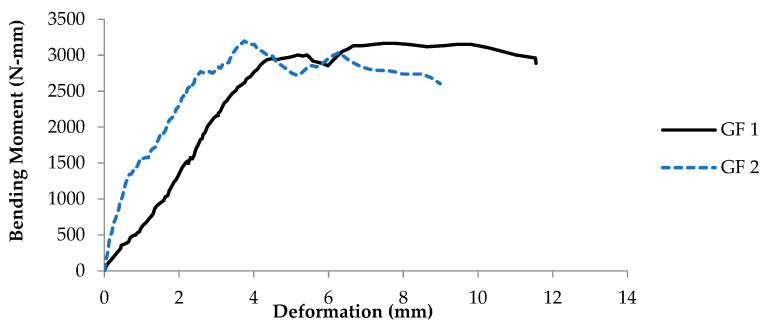
Graph between bending moment and deformation of glass fiber coated samples.

**Figure 8 materials-16-00257-f008:**
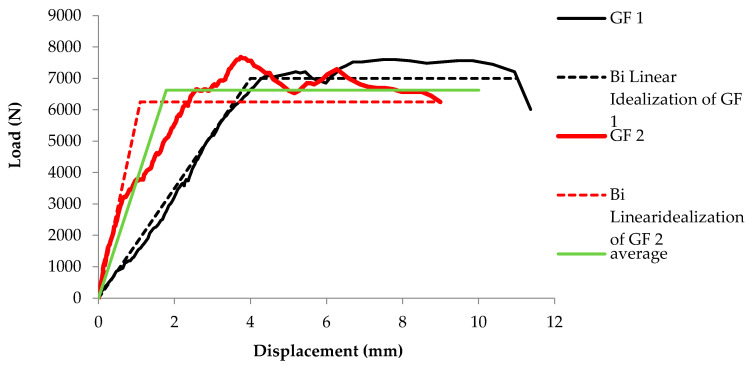
Bilinear idealized load–displacement curves for GF1 and GF2 and their mean.

**Figure 9 materials-16-00257-f009:**
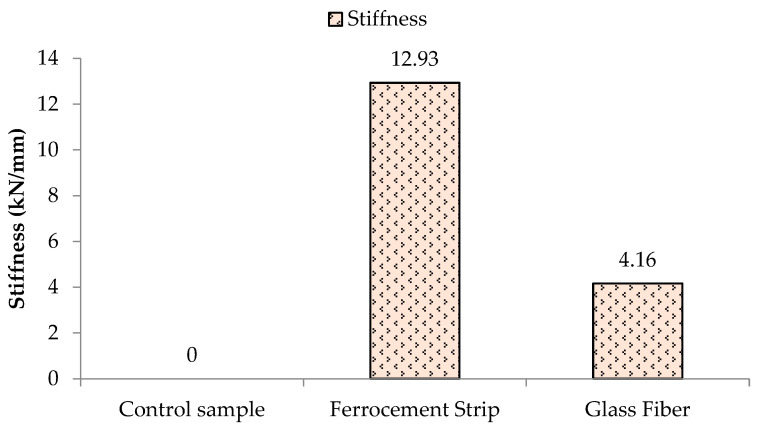
Stiffness of the unreinforced prism and retrofitted with ferrocement and glass fiber.

**Figure 10 materials-16-00257-f010:**
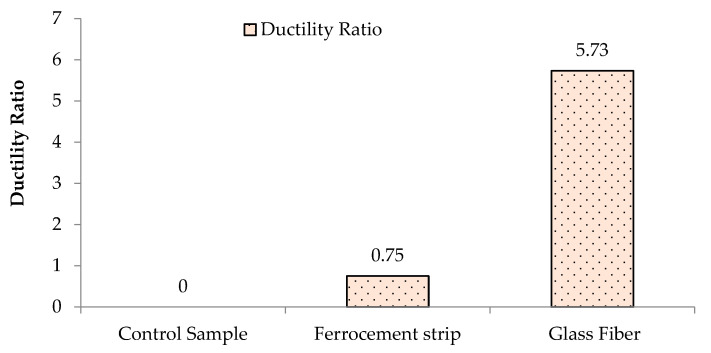
Ductility of the unreinforced prism and retrofitted with ferrocement and glass fiber.

**Figure 11 materials-16-00257-f011:**
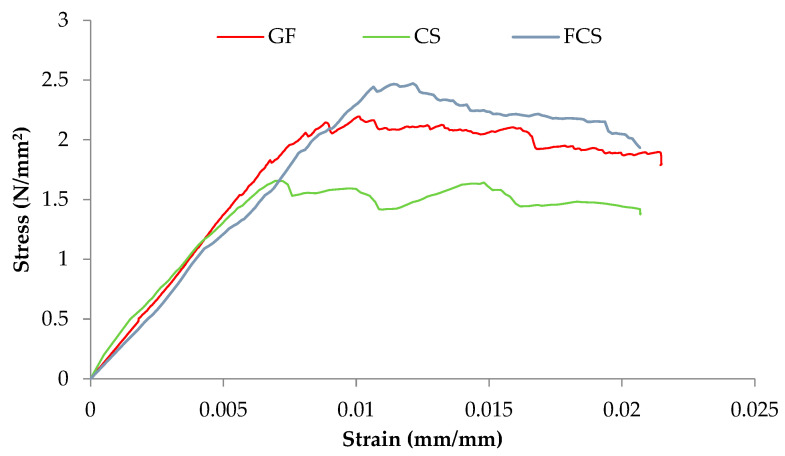
Combined stress–strain curves of compressive strength test of all prisms.

**Figure 12 materials-16-00257-f012:**
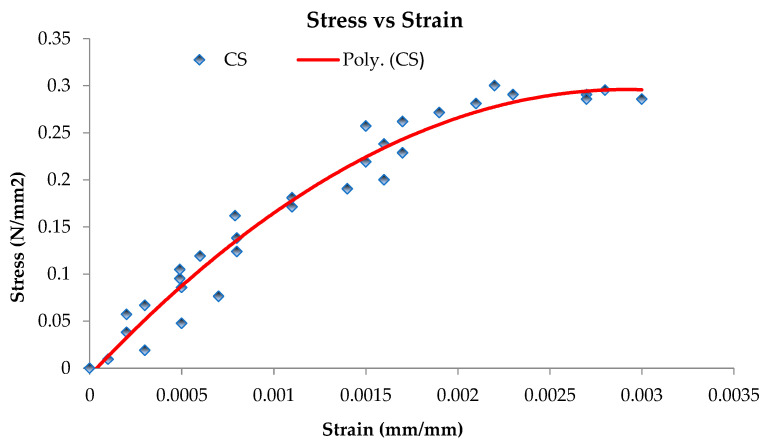
Shear stress–shear strain curve of control sample 2.

**Figure 13 materials-16-00257-f013:**
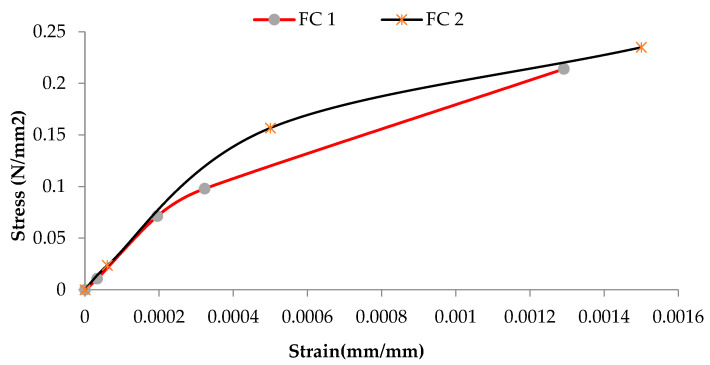
Shear stress—shear strain curve of ferrocement strip samples.

**Figure 14 materials-16-00257-f014:**
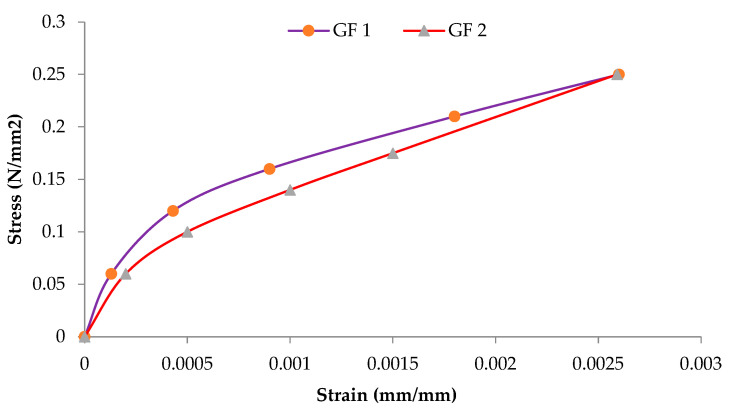
Shear stress—shear strain curve of glass fibers samples.

**Table 1 materials-16-00257-t001:** Compression capacity of three bricks taken randomly.

S. No	Max Load in kN (Tons)	Loaded Area in cm (in)	Compressive Strength in kN/m^2^ (Psi)	Ave. Strength in kN/m^2^ (Psi)
1	288.95 (29)	21.5 × 10.2	10,975.97 (1591.93)	12,064.37 (1749.79)
2	328.81 (33)	21.5 × 10.2	12,236.74 (1774.79)	
3	348.74 (35)	21.5 × 10.2	12,978.34 (1882.35)	

**Table 2 materials-16-00257-t002:** Characteristic parameters of the equivalent bilinear curves of GF1 and GF2.

Test Specimen	P_max_ (N)	P_u_ (N)	Δ_y_ (mm)	Δ_u_ (mm)	Δ_max_ (mm)
GF1	7563.34	7000	4	11	11.36
GF2	7640	6250	1.1	8.98	9

**Table 3 materials-16-00257-t003:** Compressive strength, corresponding strain, and modulus of elasticity of brick prisms.

S. No	Sample	Properties			
		Max Stress MPa (psi)	Ultimate Strain (mm/mm)	Yield Strain (mm/mm)	E in MPa (psi)
1	CS	1.657 (240.3)	0.019	0.0015	151.5 (21,968)
2	GF	2.195 (318.2)	0.021	0.0069	266.6 (38,657)
3	FCS	2.471 (358.4)	0.020	0.0044	248.3 (36,013)

**Table 4 materials-16-00257-t004:** The ultimate shear stress, strain, and shear modulus of the controlled specimen and retrofitted with glass fiber and ferrocement.

S. No	Samples	Properties		
		Max Stress in MPa (psi)	Max Strain (mm/mm)	G in MPa (psi)
1	CS	0.1815 ± 0.167 (26.31)	0.0013 ± 0.002	275 (39,875)
2	GF	0.25 (36.25)	0.0026 ±0.0001	227.3 ± 136.08 (32,967)
3	FCS	0.224 ± 0.013 (32.55)	0.0014 ±0.00021	393.86 ± 28.52 (57,110)

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
