# Peer review of "Investigating the Retrofitting Effect of Fiber-Reinforced Plastic and Steel Mesh Casting on Unreinforced Masonry Walls"

_materials, 2022, doi:10.3390/ma16010257_

Round 1
Reviewer 1 Report
This paper presents "Investigating the retrofitting effect of Fiber Reinforced Plastic and Steel Mesh Casting on the Unreinforced Masonry Walls." It is focused only on the comparison of unreinforced masonry
structures retrofitted with glass fibers and Ferro cement strips under static and dynamic loading conditions. Four tests are described - compressive strength test of brick, prism flexure test, prism diagonal tension test, and compressive strength test of prisms. The following five results are presented - brick compression strength, prism flexural strength, stiffness and ductility of prisms, the elastic modulus of prisms, and shear modulus of prisms. The expected results are then discussed in the Conclusions.
The paper written in high-level English is well structured. Graphs, tables, and figures presented illustrate the problem solved
appropriately. The theme is contemporary, and it follows up the research done in recent years, which is listed in references.
Several mistakes in the text do not reduce the quality of the paper.
For example:
Page No. 4, figure 3: bad yellow boxes with text in the left figure
Page No. 9, line No. 285: "limit.it" -> "limit. It"
Page No. 9, line No. 290: missing reference "Error! Reference source not found."
Page No. 11, figure 13: "Strtess vs Strain" -> "Stress vs Strain"
The appearance of all equations should be changed according to Journal standards.
I recommend accepting the paper for publication after a minor revision.
Author Response
The detailed reply is attached in the word file.

Author Response

(The authors gave the same response as above.)

Round 2
Reviewer 2 Report
Please refer to the Comments on materials-1918396-peer-review-v2 file attached
